# Pressures in the Ivory Tower: An Empirical Study of Burnout Scores among Nursing Faculty

**DOI:** 10.3390/ijerph20054398

**Published:** 2023-03-01

**Authors:** Sheila A. Boamah, Michael Kalu, Rosain Stennett, Emily Belita, Jasmine Travers

**Affiliations:** 1Faculty of Health Sciences, School of Nursing, McMaster University, 1280 Main Street West, Hamilton, ON L8S 4K1, Canada; 2Faculty of Health Sciences, School of Rehabilitation Science, McMaster University, 1400 Main Street West, Institute for Applied Health Science (IAHS) Building, Room 403, Hamilton, ON L8S 1C7, Canada; 3Faculty of Health Sciences, Department of Health, Evidence and Impact, McMaster University, 1280 Main Street West, Hamilton, ON L8S 4L8, Canada; 4Rory Meyers College of Nursing, New York University, 433 First Avenue, New York, NY 10016, USA

**Keywords:** burnout, emotional exhaustion, nursing faculty, workload, resilience, retention

## Abstract

(1) Background: The COVID-19 pandemic has exacerbated incidents of burnout among academics in various fields and disciplines. Although burnout has been the subject of extensive research, few studies have focused on nursing faculty. This study aimed to investigate the differences in burnout scores among nursing faculty members in Canada. (2) Method: Using a descriptive cross-sectional design, data were collected via an online survey in summer 2021 using the Maslach Burnout Inventory general survey and analyzed using the Kruskal-Wallis test. (3) Result: Faculty members (*n* = 645) with full-time employment status, worked more than 45 h, and taught 3–4 courses reported high burnout (score ≥ 3) compared to those teaching 1–2 courses. Although education levels, tenure status or rank, being on a graduate committee, or the percentage of hours dedicated to research and services were considered important personal and contextual factors, they were not associated with burnout. (4) Conclusions: Findings suggest that burnout manifests differently among faculty and at varying degrees. As such, targeted approaches based on individual and workload characteristics should be employed to address burnout and build resilience among faculty to improve retention and sustain the workforce.

## 1. Introduction

The COVID-19 pandemic has exacerbated the global nursing workforce shortages in practice and academia. According to the International Council of Nurses (ICN), as many as 13 million more nurses may be needed by 2030 [1]. The organizational and healthcare literature highlights a multitude of factors contributing to the growing workforce shortages, including an aging workforce, unhealthy work environments, heavy workload, inequitable workforce distribution, and a lack of qualified nursing faculty to train future nurses [2,3]. In academia, the dramatic increase in online course delivery and service may have intensified incidents of burnout among nursing faculty, leading to decreases in job satisfaction and high turnover [4,5,6,7]. A 2022 report has noted that the number of nurses leaving clinical practice and academia is increasing because of the stress and hardship nurses and nursing faculty continue to face throughout the COVID-19 crisis [8].

Pre-pandemic, burnout was a common phenomenon and widely prevalent in the nursing profession. However, emerging studies point to a growing concern about the rise in burnout incidences among this group since COVID-19 [4,9]. Although academics in all fields face tremendous pressures related to the work, female academics, in particular, are more susceptible to burnout and job dissatisfaction due to increased responsibilities related to traditional gender roles in child rearing and caregiving [10]. Nursing is a primarily female-dominated, practice-based profession which requires faculty to be well versed in both clinical practice as well as academic knowledge of the discipline, thus increasing their workload and subsequent risk of burnout. Nursing faculty have been presumed to experience high levels of burnout resulting from the conflicting demands of academia and clinical practice [11]. In addition to the standard academic roles and responsibilities, such as teaching, mentoring students, working on committees and research, nursing faculty are required to remain up to date with clinical practice [11]. The ability to juggle between didactic and clinical teaching especially during the pandemic, has proven to be cumbersome. Typically, nurses come into the faculty role from various professional backgrounds with different educational preparation, and also, due to differences in the nature of duties of nursing faculty (e.g., teaching-track vs. research-oriented roles), it is important to understand how these differences may predispose and/or increase faculty members’ risk of burnout. To date, the occurrence of burnout across workload and work characteristics of nursing faculty has not been empirically assessed. This study seeks to address this gap.

The aim of the present study is to investigate the differences in burnout scores across demographic (e.g., levels of education, tenure status) and workload characteristics (e.g., total hours worked, percentage of hours spent in teaching, research, service, or clinical practice). In this paper, the term ‘nursing faculty’ refers to nurses that hold academic positions at a post-secondary institution, including tenured, tenure-track, non-tenured teaching, and research streams.

### 1.1. Background

Burnout is a state of emotional, physical, and mental fatigue caused by excessive and prolonged stress [12]. Leiter et al. [12] conceptualized burnout as a triad of feelings of emotional exhaustion, depersonalization (cynicism), and a low sense of personal accomplishment (professional inefficacy) resulting from poor/negative workplace conditions, such as unmanageable workloads and lack of organizational support and resources. Emotional exhaustion is defined as feelings of being overextended and chronically fatigued by one’s work; cynicism refers to indifference or distant attitude towards work; and professional inefficacy entails feelings of reduced confidence in one’s ability or sense of competence and accomplishment in the job [12]. Burnout can manifest in the form of several different physical and emotional symptoms, such as fatigue, headache, anxiety, insomnia, reduced concentration, and forgetfulness [13], which may result in job dissatisfaction, high absenteeism, intention to leave, and job turnover [12]. Due to the severity of the impact of burnout on both employee health and work productivity, the World Health Organization in 2019 declared burnout an occupational phenomenon (a syndrome/condition), in the International Classification of Disease 11th revision (ICD-11) [14]. 

Nurses, among other healthcare professionals, have been known to have a higher prevalence of burnout symptoms because of the high physical and emotional demands of the work. A 2020 systematic review of 113 studies with a sample of 45,539 clinical nurses worldwide in 49 countries across multiple specialties reported that the overall prevalence of burnout symptoms among global nurses was 11.2% prior to COVID-19 [15]. However, a 2021 systematic review and meta-analysis including 16 studies with 18,935 staff nurses reported an overall high prevalence of emotional exhaustion, cynicism and professional inefficacy of 34.1%, 12.6%, and 15.2%, respectively [16]. Likewise, in a 2020 systematic review of 11 studies with a total of 2551 nursing faculty members which sought to identify the prevalence of burnout and associated factors reported moderate levels of burnout (59.3 out of 132) among faculty [17]. These findings highlight the prevalence of burnout among the nursing population.

Undoubtably, the COVID-19 pandemic has intensified stress and workload for nursing faculty. At the onset of the pandemic, face-to-face classes, clinical skills, and students’ clinical placements were either suspended, restricted, or restructured to an online version. These changes that occurred resulted in an increased teaching load for faculty. For instance, Yoshinaga et al. [18] reported that nursing faculty in Japan spend more time teaching than researching during pandemics compared to pre-pandemic. With additional teaching responsibility comes unanticipated burnout associated with the pandemic and the ordeal of teaching from home. Working remotely brought forth additional pressures and stress for faculty as they were saddled with both personal/family and work responsibilities and maintaining work-life balance. In a 2021 study [9] conducted in the United States (US), nursing faculty reported concerns about increased workload and change in duties or teaching structure, course requirement and course administration, and other uncertainties in the workplace, which had negative effects on their well-being and placed them at greater risk for burnout. 

Several personal (i.e., gender, level of education, race/ethnicity, marital status) and organizational factors (i.e., hours of work, number of students taught in the classroom, tenure status, full-time work, job pressure, perceived stress, collegial support, and management style) have been associated with increased burnout among nursing faculty [17,19]. Although most faculty members are overburdened with their respective duties, not all faculty experience burnout in the same way. Even within the same faculty rank or position and institutional setting, there may be individual differences in terms of how someone views, interprets, and handles burnout. Therefore, in addition to identifying causes of burnout, it is important to understand how burnout manifests among the different types/categories of nursing faculty (e.g., teaching vs. research) and the variation in burnout scores among faculty. Whereas there are copious amounts of literature on burnout among clinical nurses [16,20,21,22] and academia in general [17], very few studies have focused on burnout among nursing faculty, and the limited studies that exist were mostly conducted pre-COVID-19 and in the United States (US) [9]. As such, exploring factors associated with burnout and its core dimensions will help identify areas for intervention to reduce/minimize burnout, as it has been identified as a single reason why nursing faculty leave their job, leading to faculty shortages [11].

### 1.2. Study Hypotheses

Based on the review of theoretical literature, the hypothesized relationships to be tested in this study address the following research questions: Do total burnout scores significantly differ across individual or workload characteristics (H1) and; do emotional exhaustion, cynicism or professional efficacy burnout scale scores significantly differ across demographics of nursing faculty? (H2).

## 2. Materials and Methods

### 2.1. Study Design

A descriptive, cross-sectional survey design was used to examine the relationships among key demographic and workplace characteristics to identify the most significant factors contributing to faculty burnout. Approval from the university ethics review board (The Hamilton Integrated Research Ethics Board [HiREB—#1477]) was obtained prior to the study. Participation in this study was voluntary and responses were anonymized and reported in aggregates, in accordance with the ethics standards and guidelines.

### 2.2. Recruitment and Sample Size

We employed a convenience sampling method to recruit nursing faculty working in Canadian colleges or universities who were employed in full-time and part-time instructional/teaching and/or research positions. As an inclusion criterion, participants should have been working for at least six months at their institution. Eligible faculty members were identified via their respective university websites and sent email invitations, enclosed with the study objectives, potential risks and benefits, strategies to ensure the anonymity of responses, and a link to the questionnaire hosted on a secured online platform. Details of the study were described in a published protocol [23]. A power analysis was conducted using G*power software (v.3.1) [24] to identify the minimum sample needed for this study which yielded a total sample of 118. Thus, any sample above that number should have adequate statistical power to draw valid conclusions. 

### 2.3. Data Collection

A national survey of faculty was administered via Qualtrics software (available via https://www.qualtrics.com) between May to July 2021. Participants were provided a statement of consent on the first page of the survey. Completion of the online survey constituted each participant’s informed consent to participate. Dillman’s Tailored Design Method was followed to increase the response rate [25]. Such approaches included easy-to-answer questions and a manageable number of questions in each questionnaire; and sending email reminders to individuals who had not responded to the survey in weeks 3 and 4 after the first invitation [25]. One validated questionnaire on burnout (dependent variable) and questions regarding demographics and workload (independent variables) were used in the analyses for this paper, described below.

### 2.4. Measures

#### 2.4.1. Outcome Variables

Burnout was measured using the 16-item Maslach Burnout Inventory General Survey (MBI-GS), which consists of three sub-scales: emotional exhaustion (5 items), cynicism (5 items), and professional efficacy (6 items) [12]). Items on the MBI-GS were rated on a 7-point Likert scale ranging from 0 (never) to 6 (every day) and added together to provide a composite score for each subscale of burnout. Scores for emotional exhaustion and cynicism ranged from 0 to 30, and the score for professional efficacy ranged from 0 to 36. High scores obtained for the first two sub-scales and low for the third indicated the presence and degree of burnout (score on the efficacy sub-scale were reversed coded). In this study, a high level of emotional exhaustion (score of 27 or higher) and cynicism (13 or higher) and a low level of personal efficacy/accomplishment (9 or lower), was defined according to a normative sample of North American nurses and physicians [26] (see Table 1). Severe total burnout was defined as a mean ≥ 3.0. The MBI-GS questionnaire has been validated among nursing faculty, with a Cronbach’s alpha of 0.95 [4]. 

#### 2.4.2. Predictor Variables

Participants answered demographic questions covering educational level attainment, faculty position, tenure status, academic track, and employment status (full-time, part-time, permanent, or temporary); and workload questions, such as total hours worked, hours spent in teaching, research, service, or clinical practice, the number of graduate thesis/research committees currently serving, and the number of courses taught. We chose these independent variables based on the existing literature [17,19].

### 2.5. Data Analysis

Data collected from Qualtrics were downloaded and analyzed using STATA/IC (v17). Descriptive statistics were calculated to analyze the key study variables, including means (for continuous variables) and frequency/percentages (for non-continuous variables). All continuous data were initially checked to ensure that normality assumptions were met. Due to the sample distribution, the Kruskal-Wallis test was performed to examine the difference in each dependent variable (emotional exhaustion, cynicism, and professional efficacy scores) according to the types of specific independent variables described above. Lastly, Dunn’s pairwise Bonferroni comparison test was used post-hoc to examine the significant pairs when the Kruskal-Wallis test was significant [27].

## 3. Results

### 3.1. Demographics

A total of 1649 eligible participants were invited to complete the survey. However, 645 responded, yielding a 39.1% response rate. A little over half (52.7%) had Master’s degrees, 70.2% were full-time permanent staff, and 36.6% were either tenure-track or tenured professors. A little over 60% of the participants have worked at least six years in their current place of work. Additional description of the demographic characteristics of the sample is provided in Table 2. Faculty members reported a high degree of burnout on the MDI-GS total score (mean = 3.1; SD = 1.65), and moderate emotional exhaustion (mean = 18.41; SD = 8.41), moderate cynicism (mean score = 12.48; SD = 9.49), and moderate profession efficacy (mean score = 14.73; SD = 6.20). The section below spells out the research hypotheses that were investigated.

### 3.2. Study Hypotheses

**Hypothesis 1.** 
*Do total burnout scores significantly differ across individual or workload characteristics?*


To answer the hypothesis, the Kruskal-Wallis test was performed which indicated differences in the median value of total burnout scores between the three types of employment status: *χ*^2^(3, *N* = 645) = 15.435, *p* = 0.015. A pairwise post-hoc Dunn test with Bonferroni adjustments indicated that those with a full-time permanent status (*p* = 0.001) and a full-time temporary status (*p* = 0.007) had significantly higher total burnout score. Similarly, the Kruskal-Wallis test indicated differences in the median value of total burnout scores between the four options of the number of weekly hours worked: *χ*^2^(3, *N* = 645) = 67.152, *p* < 0.001. A pairwise post-hoc Dunn test with Bonferroni adjustments indicated that faculty who worked 45+ h had significantly higher total burnout score than those who worked 45 h or less (*p* < 0.001); and a marginal significant difference for 41–45 h vs. 35 h (*p* = 0.049). The total burnout score differs across the percentage of hours dedicated to teaching, (*χ*^2^(2, *N* = 645) = 8.926, *p* = 0.011), with the differences between 21–40% vs. 0–20% across total burnout scores (*p* = 0.004). There was a statistically significant difference between the number of course taught (1–2 courses, 3–4 courses, 5–6 course and 6+ courses) and total burnout scores (*χ*^2^(3, *N* = 645) = 21.678, *p* = 0.001). A pairwise post-hoc Dunn test with Bonferroni adjustments indicated that those teaching 6+ courses (*p* = 0.001), 5–6 courses (*p* < 0.001), and 3–4 courses (*p* = 0.093) had higher burnout scores than those teaching 1–2 courses.

**Hypothesis 2.** 
*Do emotional exhaustion, cynicism, or professional efficacy burnout scale scores significantly differ across demographics of nursing faculty?*


For both the exhaustion subscale: *χ*^2^(3, *N* = 645) = 16.548, *p* = 0.0009; the cynicism subscale: *χ*^2^(3, *N* = 645) = 13.909, *p* = 0.003; those with full time permanent (*p* = 0.0014; 0.0006, respectively), and full-time temporary (*p* = 0.0042; *p* = 0.0006, respectively) had higher burnout scores than those with part-time or in a contract position. A significant difference was noted between the number of hours worked and the emotional exhaustion sub-scale: *χ*^2^(3, *N* = 645) = 73.437, *p* < 0.001; and the cynicism sub-scale: *χ*^2^(3, *N* = 645) = 45.617, *p* < 0.001. These findings indicated those who worked 45+ h had significantly higher burnout scores than those who worked 36–40 h (*p* < 0.001) or 35 h or less. The Kruskal-Wallis test indicated differences in the median value of emotional exhaustion score (*χ*^2^(2, *N* = 645) = 6.727, *p* = 0.034); cynicism scores (*χ*^2^(2, *N* = 645) = 7.222, *p* = 0.027); and professional efficacy score (*χ*^2^(2, *N* = 645) = 6.519, *p* = 0.038); between the three options of percentage of hours dedicated to teaching. A pairwise post-hoc Dunn test with Bonferroni adjustments was only significant for 21–40% vs. 0–20% across emotional exhaustion (*p* = 0.014), cynicism (*p* = 0.011), and professional efficacy (*p* = 0.023). Significant differences were noted for the emotional exhaustion score: *χ*^2^(3, *N* = 645) = 17.472, *p* = 0.001 and the number of courses taught; but with only those teaching 6+ courses (*p* = 0.005) and 5–6 courses (*p* = 0.001) having higher burnout scores for those teaching 1–2 courses. Likewise, for the cynicism score: *χ*^2^(3, *N* = 645) = 20.932, *p* = 0.001; those teaching 6+ courses (*p* = 0.001), 5–6 courses (*p* < 0.001) or 3–4 courses (*p* = 0.002) having higher burnout scores than those teaching 1–2 courses. As for the professional efficacy score: *χ*^2^(3, *N* = 645) = 11.687, *p* = 0.008; those who taught 5–6 courses (*p* < 0.004), or 3–4 courses (*p* = 0.014) had higher burnout scores than those teaching 1–2 courses. 

There were no statistically significant differences in the median values of total burnout scores or any of the MBI-GS scales across: (a) levels of education (Bachelor’s, Master’s, and Ph.D.); (b) tenure category—clinical/teaching track, non-tenure track, and tenure/tenure-track; (c) academic rank (assistant professor, associate professor, full professor or clinical/seasonal instructor); (d) number of years worked (1 year or less, 2–5 years, 6–10 years and 10+ years), percentage of hours dedicated to research or clinics or 3–4 courses, 5–6 courses or more than 6 courses. In addition, there was no significant difference between the median score of the professional efficacy sub-scale and the four types of employment, (Appendix A).

## 4. Discussion

We set out to explore how burnout scores, either as a total score, or sub-scale scores (emotional exhaustion, cynicism, or professional efficacy/accomplishment), differed across individual and workload characteristics in the context of the COVID-19 pandemic. To the best of our knowledge, this study constitutes the first investigation of the differences in MDI-GS total burnout scores, or sub-scale scores across several demographics and workload characteristics among nursing faculty. In this study, faculty members reported a significantly high level of burnout (≥3) and moderate levels of emotional exhaustion, cynicism, and professional efficacy. A significant contribution of this study is that the findings provide a nuanced understanding of the key demographic and workload characteristics (e.g., working full-time, teaching load) that predisposes faculty to burnout and how the various dimensions of burnout manifests among this group. Our findings not only add to the organizational and burnout literature, but it extends our knowledge on how the different dimensions of burnout manifests across different groups of nurse academics. 

Consistent with the literature [10], our results indicate that nursing faculty are emotionally exhausted/overextended due to high job expectations and challenges associated with heavy workloads (teaching, research and service/scholarship), and pressure to maintain clinical competence. A study by Hosseini and colleagues [17] reported a moderate level of burnout among nursing faculty in a pre-COVID-19 systematic review, involving 11 studies conducted in various countries, including the US, Canada, Turkey, Iran, Egypt, Taiwan, and China. However, a 2022 mixed method study by Sacco and Kelly [9] found that nursing faculty in the US did not experience high levels of burnout during the height of the pandemic. Although our findings differ from those of Sacco and Kelly’s study, the differences could be attributed to the smaller sample size (*n* = 117), how burnout was measured, the types of supports and resources offered to faculty in those settings, and how the pandemic manifested in the US vs. Canada. In our study, burnout was measured using all three dimensions of the well-validated and reliable MBI tool, whereas Sacco and colleague measured burnout using a single item adapted from an a priori survey conducted by Dolan et al. [28], which had not been comprehensively validated across nursing faculty. 

Although the majority of nursing faculty in our study experienced burnout, there were differences in burnout scores across most workload variables. As expected, nursing faculty working 45 or more hours reported the highest burnout scores, followed by those working 41–45 and 36–40 h. This finding aligns with a study by Ellis [29] which reported that nurse academics routinely work 56 h per week to keep up with the administrative and teaching responsibilities, which impacted their work-life balance and contributed to burnout. Working long and excessive hours (over 48 h a week) can be detrimental to the health of workers directly and indirectly, including risk of cardiovascular diseases, depression, occupational stress, mental fatigue, sleep deprivation, and all-cause mortality [30]. The physiological and cognitive changes brought on by high workloads, such as decreased focus and attention, increased muscle tension, and coordination issues, can have a negative impact on an individual’s performance [30]. Although working long hours may not equate to productive hours, it is often a result of heavy workload. A 2021 study found that nursing faculty attributed their heavy workload to teaching more courses and larger classes without the help of teaching assistants as compared to faculty in other departments [31]. Heavy (or inequitable) workload is frequently cited as a root cause of occupational burnout. Increased burnout has been associated with job dissatisfaction and turnover intention or eventual turnover [32].

We noticed a pattern regarding differences in burnout scores and the number of courses taught. Even though the number of courses faculty members taught contributed to their workload, our findings indicated no differences in burnout scores between those who taught 3–4 courses and those who taught more than five courses. A plausible explanation for this could be attributed to the fact that burnout is a chronic condition and after a certain threshold (e.g., ≥3 courses), the experience and cumulative effect is the same. Additionally, our findings indicated differences in total burnout scores across the percentage of hours dedicated to teaching, but not to research or service/administrative duties. At the onset of the pandemic, most nursing education, clinical laboratories, and classes pivoted to online forums; therefore, nursing faculty spent more time preparing strategies to transition into the online platform without compromising the quality of education [18]. This resulted in faculty members, especially those in tenure-track and other research stream, paying less attention to research, as well as less faculty serving on graduate or research committees, as noted in our study findings. The reduction in research productivity is troubling because nursing education is evidence-based and depends on research to advance teaching strategies and inform practice that would improve patient care. Overall, our findings suggest a need for comprehensive strategies and resources to better support faculty, especially early career academics with research portfolio, to continue with their research priorities even amid an emergent event such as a pandemic. 

Surprisingly, our findings indicate no differences in total burnout scores or sub-scales across the study variables, including faculty rank (e.g., professor, associate), years of experience as a faculty member, and tenure status. These findings partially agree with the evidence on burnout among nursing faculty during COVID-19 [8,17], in that burnout is so pervasive that irrespective of one’s academic ranking or position, the negative impact is the same. We found no significant differences in burnout scores based on years of experience, which contradicts existing evidence showing that early career faculty often report high burnout levels [33]. A high level of burnout, especially emotional exhaustion from being overextended and depersonalization (a cynicism factor), has been identified as significant predictors of intention to leave academia among early career faculty [10]. It is plausible that stress responses from the COVID-19 pandemic may not be fully realized by some new/junior faculty in our study. Inconsistent with the literature [17], we found no significant differences in burnout scores among faculty members in terms of tenured versus non-tenured. Since tenure-track requirements are often linked to research and publications, most research activities were suspended during the pandemic. As a result, promotions regarding tenureship were either relapsed or postponed, removing the additional burden associated with meeting the tenure-ship requirement. Another possible explanation could be related to the uncertainty associated with the COVID-19 pandemic and the dire need to transition educational activities from face-to-face to online. Faculty members may be more concerned with fulfilling their roles and duties as educators or clinical preceptors to uphold the quality of nursing education/practice rather than focus on research. Although these are plausible reasons, there is a need to explore qualitatively faculty members’ perspectives on why non/tenure status and years of experience did not influence their burnout scores.

### 4.1. Implications

Our study confirms that burnout is prevalent among faculty members at all levels/positions and that it is largely rooted in the organizational culture and climate. Therefore, burnout must be viewed as a systemic problem requiring board-level action in order to minimize it. One of the challenges in managing burnout is that it is sometimes viewed as a personal matter or an issue with an individual rather than as a broader organizational problem. Due to the ongoing nursing workforce shortages, it is crucial for academic institutions to invest in long-term solutions and strategies aimed at addressing the organizational factors that have the biggest impact on burnout and faculty well-being. Beyond individual interventions, academic leaders should put in place a number of structural and cultural adjustments to lessen burnout, including establishing “manageable” expectations, promoting and modelling work-life integration, valuing quality work over quantity, and being aware of the current environment of constrained research funding. Leadership should concentrate on implementing significant and long-lasting adjustments to workload, including reduction in class-sizes, teaching load and the number of students enrolled in each course, and should provide help and human resource support for administrative and research tasks [31]. This is important because burnout not only negatively affects the faculty member’s well-being, it also subsequently impacts their productivity and ability to effectively teach and mentor students.

In addition to addressing the workload issues, it is paramount that academic institutions appoint effective and transformative leaders who have the capability of fostering healthy, inclusive, and safe work environments for faculty, particularly precarious academics (e.g., early career researchers (ECRs), pre-tenure, short-term contracts) to thrive. Such leaders are proactive in establishing authentic relationships with faculty and creating access to adequate support, opportunities, and resources for faculty to achieve their work goals [5] and minimize their risk of burnout [34]. Due to the current hypercompetitive academic culture, which is a major factor in the unsustainable work hours, rising burnout rates, and falling satisfaction with work-life balance, institutions should increase mechanisms of support for faculty including flexible work practices, work-life policies, easy and stigma-free access to mental health resources and reducing the emphasis on metrics or at least broadening the scope. In this period of uncertainty, academic institutions should consider adopting innovative policies and practices that promote faculty wellness such as peer support and formal and informal mentorship programs (e.g., self-selection or pairing of early career and senior faculty), as well as virtual and live counselling and other mental health services (e.g., self-help services), which may be beneficial for faculty to deal with the stressors in the workplace. Indicators of good mental health, including mindfulness, job satisfaction, work engagement, self-compassion, and resiliency, have been demonstrated to rise with the use of workplace mindfulness interventions [35]. Other psychosocial interventions (e.g., in-person and smartphone-delivered interventions) to encourage better stress management, sleep quality, and relaxation has recently attracted considerable interest as effective ways to reduce work-related burnout and enhance quality of life [35].

Given that faculty members continue to experience increased burnout after the COVID-19 pandemic, the use of constructive, emotion- and problem-focused coping techniques, which have been linked to lower burnout scores, should be taught to faculty. In this study, it is plausible that nursing faculty employed resilience as a coping strategy during pandemics, which may have enabled them to bounce back from unexpected circumstances through a dynamic adaptation process that results in positive coping, control, and integration [36,37]. Resilience has been positively associated with quality of life and could reduce burnout among nursing faculty [7]. In contrast to popular belief, resilience, well-being, and optimism are different aspects of mental health that are associated with enjoyment and optimal performance rather than merely the absence of distress or suffering [35]. Therefore, cultivating resilience skills (e.g., mindfulness-based resilience training) among faculty members could be beneficial not only for ensuring their wellness, but also for improving organizational performance, retention and for maintaining a healthy workforce.

### 4.2. Strength and Limitations

The strength of this study lies in the large sample size and the recruitment of nursing faculty across Canada, highlighting the broader implication of the findings. This is the first nationwide study to explore associated factors in burnout scores among nursing faculty, providing a unique perspective to the ongoing discourse on burnout and the global nursing workforce shortage. The limitations of the study, however, include the cross-sectional design which limits the generalization of the findings. Additionally, this paper did not include the exposure to COVID-19 as an independent variable and as a result we were not able to measure its impact on level of burnout among faculty. 

Future research should consider using longitudinal designs to explore a trajectory pattern of burnout across the continuum period of COVID-19, as most nursing programs are gradually returning to in-person classes. Further, the literature alludes to higher rates of burnout, fewer promotional opportunities, and lower rates of tenure among minority faculty members and female academic compared to male (white) academics [10,19]. As such, future studies should consider utilizing an intersectional perspective to examine the role that gender, race and ethnicity play across the different aspects of the job among faculty of varying academic ranks and fields. Forthcoming studies with a large dataset allowing for several interaction effects of key demographic factors, such as rank, faculty position, working conditions, and their combined impact on burnout and productivity, could provide insight into the myriad factors influencing nurse burnout and offer tailored interventions to mitigate its harmful effects.

## 5. Conclusions

Arguably, burnout is the biggest public health crisis of the 21st century as it negatively impacts many industries, including healthcare and education. This study sheds light on the prevalence of burnout across work and workload characteristics among nursing faculty. Although most faculty members in this study reported high levels of burnout and emotional exhaustion, our findings suggest that burnout manifested differently among faculty and at varying degrees. As such, targeted approaches based on individual and workload characteristics should be employed to address burnout and build resilience among faculty to improve retention. As nursing continues to face workforce shortages, immense effort must be devoted to creating healthy work environment and reducing burnout including establishing equitable policies that direct and guide workload management and promote work-life balance among faculty.

## Figures and Tables

**Table 1 ijerph-20-04398-t001:** Response categories on Maslach Burnout Inventory Survey (MBI-GS).

Response Category	Emotional Exhaustion	Depersonalization (Cynicism)	Professional Efficacy (or Accomplishment) *
Low (mild)	0–16	0–6	<9
Moderate	17–26	7–12	10-16
High (severe)	≥27	≥13	≥17

*Note*. * Score is reversed for personal efficacy/accomplishment.

**Table 2 ijerph-20-04398-t002:** Demographic characteristics of the sample.

Characteristic	*n* (%)
*Age*	
≤39 years	145 (22.5)
40–49 years	191 (29.6)
50–59 years	195 (30.2)
≥60 years	106 (16.4)
Prefer not to say	8 (1.2)
*Sex*	
Female	604 (93.6)
Male	36 (5.6)
Other	5 (0.8)
*Marital Status*	
Single	79 (12.2)
Married	441 (68.4)
Living common law	68 (10.5)
Divorced/separated	50 (7.8)
Widowed	7 (1.1)
*Highest education*	
PhD	220 (34.1)
Masters	340 (52.7)
Bachelors	79 (12.3)
Diploma	6 (0.9)
*Academic rank*	
Lecturer	82 (12.7)
Assistant Professor	144 (22.3)
Associate Professor	230 (35.7)
Full Professor	88 (13.6)
Clinical/Sessional Instructor	101 (15.7)
*Tenure status*	
Tenured	152 (23.6)
Tenure track	82 (12.7)
Teaching track	168 (26.0)
Non-tenure track	149 (23.1)
Clinical track	92 (14.3)
*Employment status*	
Full-time permanent	453 (70.2)
Full-time temporary	75 (11.6)
Part-time	117 (18.1)
*Years worked at current organization*	
≤1 year	45 (7.0)
2–5 years	200 (31.0)
6–10 years	136 (21.1)
≥10 years	264 (40.9)
*Hours worked per week*	
≤35 h	86 (13.3)
36–40 h	121 (18.8)
40–45 h	119 (18.4)
≥46 h	319 (49.5)
*Number of courses taught per year*	
1–2 courses	96 (14.2)
3–4 courses	262 (38.8)
5–6 courses	173 (25.6)
>6 courses	114 (16.9)
*Institutional location (by region)*	
Central Canada	212 (32.9)
The Prairie Provinces	201 (31.2)
The West Coast (Pacific Region)	133 (20.6)
The Atlantic Region	93 (14.4)
The Northern Territories	6 (0.9)
*Institution type*	
University	524 (81.2)
College	121 (18.8)
*Institution size*	
Small	185 (28.7)
Mid-size	215 (33.3)
Large	245 (38.0)

## Data Availability

Research data are not publicly available due to restrictions (e.g., contains information that could compromise the privacy of research participants), and in accordance with the ethics agreement. However, data can be obtained from the principal investigator upon reasonable request.

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
