# Peer review of "Pressures in the Ivory Tower: An Empirical Study of Burnout Scores among Nursing Faculty"

_ijerph, 2023, doi:10.3390/ijerph20054398_

Round 1

Reviewer 1 Report

Dear Authors,

Thank you for the opportunity to review this interesting work. The topic of Burnout is particularly relevant, especially related to health care professionals who have been under great psycho-physical strain since before covid19. Among the variables considered, your results showed that the level of burnout is particularly related to employment status, hours worked, and number of courses taught. Although these important results with several practical implications, the paper lacks in some parts and insights.

INTRODUCTION

I think it is important to explain the purpose of the research from the outset. In particular, I would suggest moving the final part of the theoretical background (pg 3, line 127-135) to the end of the introduction paragraph. As far as the background is concerned, I think it is necessary to argue each variable that one has decided to include on the basis of the literature and also add specific hypotheses, which will then guide the description of the results and the discussion.

METHOD AND MATERIALS

All the information are well explained.

RESULTS

Table 2 is not formatted; it looks like it was copied and pasted from an output. It should be formatted in line with the journal's requirements.

The results should then be reported in the light of the hypotheses made.

To lighten the manuscript, it is possible to describe only the positive results and perhaps add tables with all the variables in the supplementary materials.

Was covid19 also considered among the variables? E.g., whether the participants contracted the virus, or if the participants also worked in covid19 wards. It is not clear to me whether the participants were only doing academics or were also working professionals in the wards.

DISCUSSION

The discussion should follow the assumptions made in the introduction, commenting on both significant and non-significant results. For instance, why tenure status is not significant related to burnout?

Resilience would have been interesting to analyse, as a psychological variable. Let us say that psycho-social variables are absent from this study and perhaps would have allowed for a better commentary.

Practical implication

It could be added.

I think it is important to add other practical implications: what are these data for? How can managers, directors, use them to reduce burnout risk, using your results? What interventions can be made? E.g., can digital interventions be envisaged? See review by Paganin & Simbula. 2020, in which two studies aimed at nurses are reported, one of which is to increase resilience (Mistretta et al.,2018).

Study limitations and future direction

They could be sufficient.

CONCLUSION

The conclusions summarise the results and broadly identify the usefulness of the study. The latter can be emphasised, especially in light of the high prevalence of Burnout among nurses and the negative outcomes of this spread.

Author Response

Response to Reviewer 1’s Comments

Dear Authors,

Thank you for the opportunity to review this interesting work. The topic of Burnout is particularly relevant, especially related to health care professionals who have been under great psycho- physical strain since before covid19. Among the variables considered, your results showed that the level of burnout is particularly related to employment status, hours worked, and number of courses taught. Although these important results with several practical implications, the paper lacks in some parts and insights.

Comment 1

INTRODUCTION

I think it is important to explain the purpose of the research from the outset. In particular, I would suggest moving the final part of the theoretical background (pg 3, line 127-135) to the end of the introduction paragraph. As far as the background is concerned, I think it is necessary to argue each variable that one has decided to include on the basis of the literature and also add specific hypotheses, which will then guide the description of the results and the discussion.

Response 1

Thank you for the comment and suggestion. The manuscript has been revised.

Please see page 2, paragraph 1 – “The aim of the present study is to investigate the differences in burnout scores across individual (e.g., levels of education, tenure status) and workload characteristics (e.g., total hours worked, percentage of hours spent in teaching, research, service, or clinical practice)”

On page 2, paragraph 3 of the manuscript, we defined the dependent variable. The independent variables on the other hand are not well-defined concepts per se that requires formal definitions (i.e., educational level, employment status etc.), however, we have, provided justifications for the demographic variables used in the paper, please see page 3, paragraph 2  (see lines 117-133). We have also added a statement at the method section to highlight this (see please see page 4, paragraph 4, lines 193-194). As well, we have provided additional details in the tables.

We have added hypotheses to along with the research questions. Please see 3, paragraph 3 (lines 135-139).

Comment 2

METHOD AND MATERIALS

All the information are well explained.

Response 2

Thank you for the comment.

Comment 3

RESULTS

Table 2 is not formatted; it looks like it was copied and pasted from an output. It should be formatted in line with the journal's requirements.

Response 3

Thank you for the comment. Table 2 has been re-formatted to align with Journal’s requirement. Please see page 5, paragraph  2 and page 6.

Comment 4

The results should then be reported in the light of the hypotheses made.

Response 4

Thank you for the comment. We have reported the result based on the hypotheses made. Please see page 6, paragraph 1 and page 7, paragraphs 1-3.

Comment 5

To lighten the manuscript, it is possible to describe only the positive results and perhaps add tables with all the variables in the supplementary materials.

Response 5

Thank you for the comment. It is important for the reader to get a full scope of the analysis both significant and non-significant findings. However, we have reported only the positive result (see page 7, paragraphs 3), and have referred the reader to the Supplementary material-Table S1 online.

Comment 6

Was covid19 also considered among the variables? E.g., whether the participants contracted the virus, or if the participants also worked in covid19 wards. It is not clear to me whether the participants were only doing academics or were also working professionals in the wards.

Response 6

Thank you for the comment. Our study did not include exposure to COVID-19 as an independent variable our study and as such, we have now stated this as a limitation.

Please see page 10, paragraph 3 – “The limitations of the study, however, include the cross-sectional design which limits the generalization of the findings. Additionally, this study did not include the exposure to COVID-19 as an independent variable and as a result we were not able to measure its impact on level of burnout among nursing faculty.” We suggested that “future research should consider using longitudinal designs to explore a trajectory pattern of burnout across the continuum period of COVID-19, as most nursing programs are gradually returning to in-person classes.”

Due to the pandemic, it is possible that some returned to assist clinically on a casual basis, but the data only reflected what they shared.

Comment 7

DISCUSSION

The discussion should follow the assumptions made in the introduction, commenting on both significant and non-significant results. For instance, why tenure status is not significant related to burnout?

Response 7

Thank you for the comment. The Discussion did focus on both significant and non-significant findings. We have added a paragraph to highlight the non-significant findings of tenured status. Please see page 9, paragraph 1 (see lines 342-365).

Comment 8

Resilience would have been interesting to analyse, as a psychological variable. Let us say that psycho-social variables are absent from this study and perhaps would have allowed for a better commentary.

Response 8

Thank you for the comment. Yes, resilience would have been a good psychological variable to measure in this paper. Due to the length of the study questionnaire, some of these variable could not be included. We were cognisant of no overburdening the participants as we are ironically measuring workload and burnout. It is our hope to conduct a follow up study to capture some of these other interesting concepts. We have included some discussions on resilience in the revised manuscript. Please see page 10, paragraph 2.

Comment 9

Practical implication

It could be added.

Response 9

Thank you. Please see page 9, paragraph 2 and page 10, paragraph 1.

Comment 10

I think it is important to add other practical implications: what are these data for? How can managers, directors, use them to reduce burnout risk, using your results? What interventions can be made? E.g., can digital interventions be envisaged? See review by Paganin & Simbula. 2020, in which two studies aimed at nurses are reported, one of which is to increase resilience (Mistretta et al.,2018).

Response 10

Thank you for the comment and the resource. Please see page 9, paragraph 2.

Comment 11

Study limitations and future direction

They could be sufficient.

Response 11

Thank you for the comment.

Comment 12

CONCLUSION

The conclusions summarise the results and broadly identify the usefulness of the study. The latter can be emphasised, especially in light of the high prevalence of Burnout among nurses and the negative outcomes of this spread.

Response 12

Thank you. Please see page 10, paragraph 4, and page 11, paragraph 1.

Reviewer 2 Report

This is a well-structured paper reporting on a topic of timely significance. All aspects of COVID-19's impact on teaching, training, and care delivery across the healthcare professions need to be documented, and this paper makes an important contribution by looking at associations between burnout and various aspects of workload and work characteristics in a national sample of Canadian nursing faculty. The paper engages well with the literature and adds findings that those interested in burnout in the health professions will find informative. 

I have some minor critiques:

- The prominent use of the word "demographics" in the title, and throughout the paper, led me to expect that the paper would look at sex/gender, race/ethnicity, and age as moderating/mediating factors, and I suspect many readers would have the same expectation, especially given the way the introduction acknowledges the importance of gender and race/ethnicity in burnout. It is a limitation of the paper that these factors were not considered in the analyses. It's not clear why race/ethnicity data were not collected; perhaps less importantly, it's also not clear why age group (as distinct from years in profession) was not looked at, when the survey collected these data. The respondents were overwhelmingly women which I assume led to a lack of power to look at sex/gender differences, but this could be stated explicitly. I would look for ways to emphasize "workload" or "work characteristics" in place of "demographics" in the text. To give an example, the sentence that reads "Surprisingly, our findings indicate no differences in total burnout scores or sub-scales 349 across most demographic variables" (line 349, my emphasis) somehow actively reminds me of all the common variables not looked at, and would perhaps be better written as something like "across the variables included in this study."

- One of the more interesting findings, to me, was that there was not a difference in burnout between nurses in clinical and other, more research-oriented roles. I think many people assume that burnout was greater for those engaged in direct patient care throughout the pandemic, but in fact it may be that the directly observable rewards of patient care may have been, in some sense, protective against burnout, at least in the early stages of the pandemic. The authors don't need to speculate on this, but I would suggest highlighting this finding a little more in the discussion - as it is, it's easily missed.

- The response rate to the survey is acceptable but is a little low, and this should be acknowledged in the limitations. 

- In lines 323-327,  the authors suggest that resilience skills might provide an explanation for why burnout did not continue to increase with each additional course taught by faculty nurses. I agree resilience is an attribute/skill that mitigates burnout, but I couldn't quite follow why it would especially or only come into play in relation to the number of courses taught, so this argument was unclear to me.  

Author Response

Response to Reviewer 2’s Comments

This is a well-structured paper reporting on a topic of timely significance. All aspects of COVID- 19's impact on teaching, training, and care delivery across the healthcare professions need to be documented, and this paper makes an important contribution by looking at associations between burnout and various aspects of workload and work characteristics in a national sample of Canadian nursing faculty. The paper engages well with the literature and adds findings that those interested in burnout in the health professions will find informative.

I have some minor critiques:

Comment 1

- The prominent use of the word "demographics" in the title, and throughout the paper, led me to expect that the paper would look at sex/gender, race/ethnicity, and age as moderating/mediating factors, and I suspect many readers would have the same expectation, especially given the way the introduction acknowledges the importance of gender and race/ethnicity in burnout. It is a limitation of the paper that these factors were not considered in the analyses. It's not clear why race/ethnicity data were not collected; perhaps less importantly, it's also not clear why age group (as distinct from years in profession) was not looked at, when the survey collected these data. The respondents were overwhelmingly women which I assume led to a lack of power to look at sex/gender differences, but this could be stated explicitly. I would look for ways to emphasize "workload" or "work characteristics" in place of "demographics" in the text. To give an example, the sentence that reads "Surprisingly, our findings indicate no differences in total burnout scores or sub-scales 349 across most demographic variables" (line 349, my emphasis) somehow actively reminds me of all the common variables not looked at, and would perhaps be better written as something like "across the variables included in this study."

Response 1

Thank you for the comment. The entire manuscript has been thoroughly revised accordingly to address your comments. As for the last sentence, changes are reflected on page 9, paragraph 1.

Also, the title of the manuscript has been changed to, “Pressures in the ivory tower: An empirical study of burnout scores among nursing faculty.”

Comment 2

- One of the more interesting findings, to me, was that there was not a difference in burnout between nurses in clinical and other, more research-oriented roles. I think many people assume that burnout was greater for those engaged in direct patient care throughout the pandemic, but in fact it may be that the directly observable rewards of patient care may have been, in some sense, protective against burnout, at least in the early stages of the pandemic. The authors don't need to speculate on this, but I would suggest highlighting this finding a little more in the discussion - as it is, it's easily missed.

Response 2

Thank you for this insight. We have highlighted devoted a paragraph to discuss why we think the burnout scores did not differ across tenure status (including those In clinical roles). Please see page 9, paragraph 1.

Comment 3

- The response rate to the survey is acceptable but is a little low, and this should be acknowledged in the limitations.

Response 3

Thank you for the comment. For the type of sample and the subject matter, the sample size is moderate and higher than what has been reported in the nursing literature. In another published article, noting the sample size of 645 as a limitation was discouraged by the reviewers and editor. In this study, a power analysis was conducted using G*power software to identify the minimum sample needed for this study which yielded a total sample of 118. Therefore, any sample above that number have adequate statistical power to draw valid conclusions. Please see page 4, paragraph 1.

Comment 4

- In lines 323-327, the authors suggest that resilience skills might provide an explanation for why burnout did not continue to increase with each additional course taught by faculty nurses. I agree resilience is an attribute/skill that mitigates burnout, but I couldn't quite follow why it would especially or only come into play in relation to the number of courses taught, so this argument was unclear to me.

Response 4

Thank you. Point well taken and revisions have been made accordingly. Please see page 9, paragraph 3.